# Multi-stage feature selection (MSFS) algorithm for UWB-based early breast cancer size prediction

V. Vijayasarveswari[1], A. M. Andrew[1]*, M. Jusoh[1], T. Sabapathy[1], R. A. A. Raof[1], M. N. M. Yasin[1], R. B. Ahmad[1], S. Khatun[2], H. A. Rahim[1]

1 Advanced Communication Engineering (ACE) Centre of Excellence, Universiti Malaysia Perlis, Kangar, Perlis, West Malaysia, 2 Faculty of Electrical & Electronic Engineering, Universiti Malaysia Pahang, Pekan, Pahang

☯ These authors contributed equally to this work.
* allanmelvin@unimap.edu.my

**Data Availability Statement:** The authors have published the minimal dataset called 'EBCD_MSFS Dataset' in figshare. The DOI ID is http://dx.doi.org/10.6084/m9.figshare.12746546.

## Abstract

Breast cancer is the most common cancer among women and it is one of the main causes of death for women worldwide. To attain an optimum medical treatment for breast cancer, an early breast cancer detection is crucial. This paper proposes a multi- stage feature selection method that extracts statistically significant features for breast cancer size detection using proposed data normalization techniques. Ultra-wideband (UWB) signals, controlled using microcontroller are transmitted via an antenna from one end of the breast phantom and are received on the other end. These ultra-wideband analogue signals are represented in both time and frequency domain. The preprocessed digital data is passed to the proposed multi- stage feature selection algorithm. This algorithm has four selection stages. It comprises of data normalization methods, feature extraction, data dimensional reduction and feature fusion. The output data is fused together to form the proposed datasets, namely, 8-HybridFeature, 9-HybridFeature and 10-HybridFeature datasets. The classification performance of these datasets is tested using the Support Vector Machine, Probabilistic Neural Network and Naïve Bayes classifiers for breast cancer size classification. The research findings indicate that the 8-HybridFeature dataset performs better in comparison to the other two datasets. For the 8-HybridFeature dataset, the Naïve Bayes classifier (91.98%) outperformed the Support Vector Machine (90.44%) and Probabilistic Neural Network (80.05%) classifiers in terms of classification accuracy. The finalized method is tested and visualized in the MATLAB based 2D and 3D environment.

## Introduction

The rate of a woman contracting breast cancer is reported at a worrying rate globally, particularly in developing countries. Symptoms of breast cancer, for instance, visual changes in the breasts are usually discovered only at the final stage [1]. Consequently, most of the breast

**Funding:** The study was supported by a grant from Ministry of Education, Malaysia: FRGS – 9003-00418. The funders had no role in study design, data collection and analysis, decision to publish, or preparation of the manuscript.

**Competing interests:** The authors have declared that no competing interests exist.

cancer cases are detected in the latter stage, at which, are deemed as too late for medical treatment, thus causing death [1, 2].

Malaysian National Cancer Registry (NCR) Report published every 5 years recorded that the breast cancer is the most common cancer type, holding the top position out of the other common cancer type [3]. The report also states that the Age- Standardized Incidence Rate (ASR) for female is 34.1 per 100 000 populations in the year 2012-2016. Age- Standardized Incidence Rate for male is also recorded, at the increased rate of 0.5 per 100 000 population [3].

In another report generated by GLOBOCAN in 2018, states that breast cancer holds record as second most commonly diagnosed cancer type in the world, with 2.089 million incidences of reported new cases (11.6%) [4]. Based on the reports, it can be clearly concluded that breast cancer cases are increasing every year and it is still recorded as second top causes of the woman's death [4, 5].

There are many existing clinical methods in diagnosing and detecting breast cancers. Common diagnostic methods are mammography, magnetic resonance imaging (MRI) and ultrasound scans [6–7]. However, these methods are proven costly, bulky, invasive and are unable to detect the early stages of breast cancer. These limitations are the main barriers for an efficient early breast cancer detection. Detection of breast cancer in the early stage is very crucial for further medical diagnostics and treatment. Slow detection is indirectly reducing the survival rate of the patients [6].

Taking into consideration all the limitations of the conventional diagnostic methods, microwave based ultra-wide-band (UWB) imaging technology can be a potential and promising method for early breast cancer detection as it is convenient, non- invasive, secure and low-cost [7, 10–12]. The research involving UWB imaging for breast cancer started effectively in year 1999 by S.C. Hagness from Winconsin University, USA [13], and since then, started to gain popularity among the researchers, credit to the advancement of computational power since late 1990s. Basically, researchers used either real-time machines such as a vector network analyzer or machine learning to analyze the UWB signals [1].

Talking about the analysis of UWB signals using machine learning methods, the capabilities of the classifier model are dependent on the features fed into the machine learning model for the training purpose. Better is the feature, higher will be the classification rate of the classifier model. In general, the features are selected based on the different feature selection methods proposed by various researchers in breast cancer size detection. Researchers normally identify their features by mean of feature extraction, feature selection or feature dimensionally reduction methods [1]. Previous researchers depict use of conventional feature selection method, basically, by using a single- stage feature selection method [1]. In single- stage feature selection method, the important features are extracted from the raw data, and the extracted data is further filtered to select only important and useful features. The method is having some drawback where there can be misclassification due to the deficiency of quality data during the feature extraction stage. The exploration and exploitation of the data will be insufficient during the feature selection as the features are reduced at the initial stage. As a result, only some redundant features are selected, and some useful features are lost due to poor data management [14, 15].

The proposed Multi- Stage Feature Selection (MSFS) can be a solution in overcoming the mentioned drawbacks of single- stage feature selection method. MSFS can increase the learning model performance in breast cancer detection application [14, 15]. The proposed multi-stage feature selection method will be discussed comprehensively throughout the paper. The performance of the proposed method is validated using statistical and machine learning approaches.

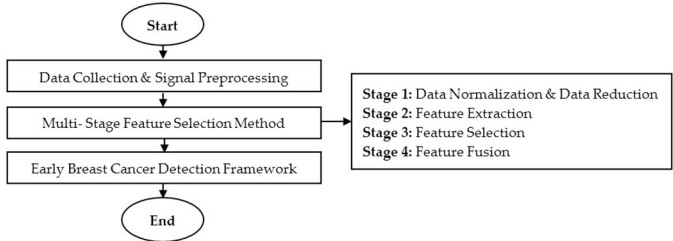

**Fig 1. The overall experimental process.** Figure shows the flowchart of the overall experimental process involved in this research.

## Materials and methods

In this section, the breast cancer sampling technique, the feature extraction from UWB sensors using various data normalization techniques, the proposed feature selection algorithm, and the classification stages are explained.

Fig 1 shows the flowchart of the overall experimental process involved in this research. The process started with data collection using breast phantoms, and signal preprocessing. Then, it will be followed by the proposed MSFS method which comprises of four stages.

In the first stage, the preprocessed data is normalized using various data normalization method, and the data is reduced using Principal Component Analysis (PCA). Data normalization process is important since it is important to select the best features without eliminating useful information from the preprocessed data.

The second stage and all the subsequent stages comprise of feature extraction, feature selection based on the statistical approach, and feature fusion to form the proposed feature that will be incorporated into the Early Breast Cancer Detection (EBCD) Framework. The early breast cancer detection will be visualized in 2D and 3D environment.

## Data collection

The data collection is conducted using breast phantoms. The breast phantoms have been developed using different materials [16–20]. It is important to make sure that the breast phantoms are having comparable real breast's dielectric properties in terms of permittivity and conductivity, mimicking the real breast tissue. Based on the literature studies conducted [16–20], most of the researchers use low- cost and non- chemical ingredients like Vaseline (petroleum jelly), a mixture of wheat- flour, water, and soy oil to develop heterogeneous breast phantoms.

The breast phantoms used in this research adopted the same model suggested by the previous researchers [18–20]. Hemispherical wine glass with 75 mm width, 60 mm height, and 1.9 mm thickness is used as a breast phantom skin. The heterogeneous breast phantom is developed using 100:50:37 ratio of the mixture of petroleum jelly, soy oil, and wheat flour. 25% water is also added to the mixture. Tumors are developed using 10:5.5 mixture ratio of water to the wheat flour. Different tumor sizes are developed for testing (2 mm, 3 mm, 4 mm, 5 mm, and 6 mm). Fig 2 shows the developed breast phantom and tumor for the experiments.

Fig 3 shows the experimental set-up for the breast cancer sampling [18, 21–24]. A pair of antennae is placed facing each other with the breast phantom located at the middle of the antennae as shown in Fig 4. Feeding cables are used to connect the UWB transceivers with antennae. The UWB signals are generated by the UWB transceivers, passed to transmitter antenna to transmit it on one end and received by the receiver antenna at the diagonal opposite end, concurrently. The captured forward scattered UWB signals are passed to UWB

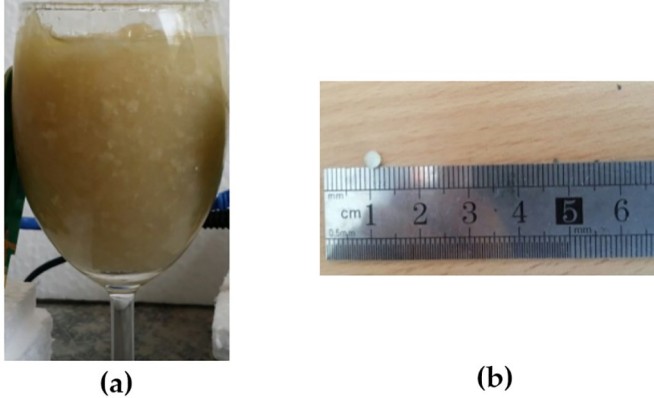

**(a)**  **(b)**

**Fig 2.** The Developed (a) Breast Phantom (b) Tumor. Figure shows the developed breast phantom and tumor for the experiments.

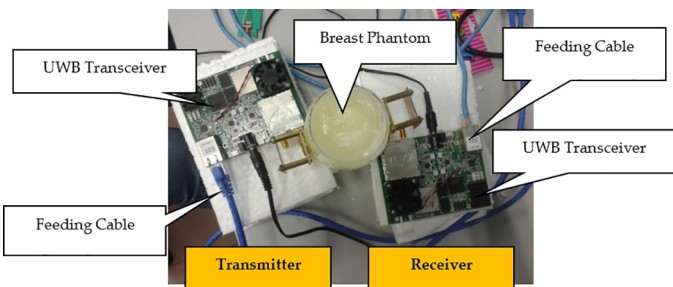

**Fig 3. Experimental set-up for breast cancer sampling.** Figure shows the experimental set-up for the breast cancer sampling.

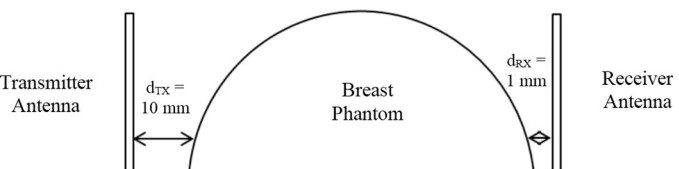

**Fig 4. The measurement setup of transmitter and receiver UWB antenna.** Figure shows the measurement setup of the transmitter and receiver UWB antenna.

transceiver in the other end. Table 1 shows the dielectric properties of breast phantom and tumor used in this research, mimicking the dielectric properties of real human breast. The experimental setup used in this work is similar to the approach used in works discussed in [25] and [26]. Such setup has shown that certain type of containers are still being used to hold the breast phantom during the measurement. Furthermore, the results are normalized after the measurement are carried out. Therefore, the containers permittivity, either glass or any other materials will have different level of signal strength, but after normalization, it should be the same. Thus, the proposed framework can work perfectly with the effect with or without glass. Similar approach of normalization is also performed in signal baseline drift correction in [14].

**Table 1. Dielectric properties of breast phantom and tumor. [18–20].**

| Breast Phantom | Material | Permittivity | Conductivity (S/m) |
|---|---|---|---|
| FattyTissue | Pure petroleum jelly | 2.36 | 0.012 |
| Glandular | Mixture of water and wheat flour | 6.98 | 0.785 |
| Glandular | Soy oil | 2.7 | 0.061 |
| Tumor | Mixture of water and wheat flour | 6.98 | 0.785 |

Table shows the dielectric properties of breast phantom and tumor used in the data collection.

The antennae achieved 6.09 dB gain and 8.15 dBi directivity during the antenna simulation [27, 28]. They are placed close to the breast phantom to avoid any loss of signals and to reduce noises. The UWB transceivers with frequency range of 3 GHz to 10 GHz are used. They are connected to the MATLAB software through Ethernet cross connectors (feeding cables). The receiver antenna captured the forward-scattered signals at the center frequency of 4.3 GHz. Table 2 shows the description of UWB patch antenna used in this research. The detailed information of the antenna, such as layout, S11 analysis, as well as the other related details could be found in work [27].

The data collection steps are as follow [18]:

Step 1: The 2 mm tumor is placed at starting location in the breast phantom.

Step 2: UWB signals are transmitted by antenna and forward scattered UWB signals are captured by the opposite antenna. 50 repetitions are taken at one point.

**Table 2. Description of UWB patch antenna used in this research. [27, 28].**

| Properties | Parameters |
|---|---|
| Type | Non- wearable |
| Dimension | 3D |
| Frequency Range(GHz) | 3.25 to 12 |
| Centre Frequency(GHz) | 4.3 |
| Dielectric Substrate(Fabrication) | I shaped FR4 |
| Dielectric Substrate(Description) | Thickness: 1.6 mm |
| | Dielectric constant: 4.3 |
| | Loss tangent: 0.025 |
| Patch(Fabrication) | Rectangular shaped FR4 |
| Patch Description | Thickness: 1.6 mm |
| | Dielectric constant: 4.3 |
| | Loss tangent: 0.025 |
| Reflector | Dimension: 60 x 45 x 0.01 $mm^3$ |
| | Location: 12.8 mm from ground plane (Back side) |
| Fabrication Measurement(mm) | Patch |
| | Width: 19 |
| | Length: 13.8 |
| | Thickness: 0.01 |
| | Dielectric Substrate: 30 x 26 x 1.6 $mm^3$ |
| Reflection co–efficientvalue, S11(dB) | -62.5 |
| Gain(dB) | 6.09 |
| Directivity(dBi) | 8.15 |

Table shows the description of UWB patch antenna used in this research.

Step 3: The tumor is placed at 27 different locations within the breast phantom. Each tumor (of same size) is placed at different location using the combination location of x coordinate (0.25 cm, 2 cm, 3.25 cm, 5 cm and 6.25 cm), y coordinate (0.25 cm, 2 cm, 3.25 cm, 5 cm and 6.25 cm) and z coordinate (3 cm, 4 cm, 5 cm).

Step 4: Step 1 to Step 3 are repeated until all the locations in the breast phantom are covered. The tumor size is then changed. Step 1 to Step 4 are repeated until the UWB signals are captured for all five different tumor sizes. A total of 6750 UWB signals are collected. Each signal sample has 1632 data points. A sample of forward scattered time domain signals (transmitted and received) are shown in Fig 5.

In general, the signal exists in time domain. It is easier to visualize the signal characteristics in time domain. However, analyzing the signal characterization in frequency domain is equally important because it helps to observe the characteristics of the signal which are unable to be visualized in the time domain [27, 29, 30]. Thus, the time domain signals obtained from the UWB transceivers are transformed to the frequency domain signals using the commonly used Fast Fourier Transform (FFT). Fig 6 demonstrates the representation of the signal in the frequency domain after the transformation. The highest peak of the signal is approximately at 4.3 GHz same as the center frequency of the UWB antenna used.

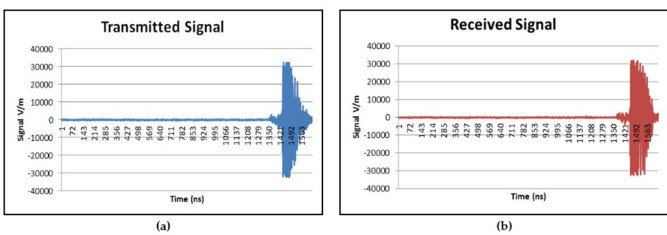

**Fig 5.** (a) Transmitted UWB Signal and (b) Received UWB Signal. Figure shows a sample of forward scattered time domain signals (transmitted and received).

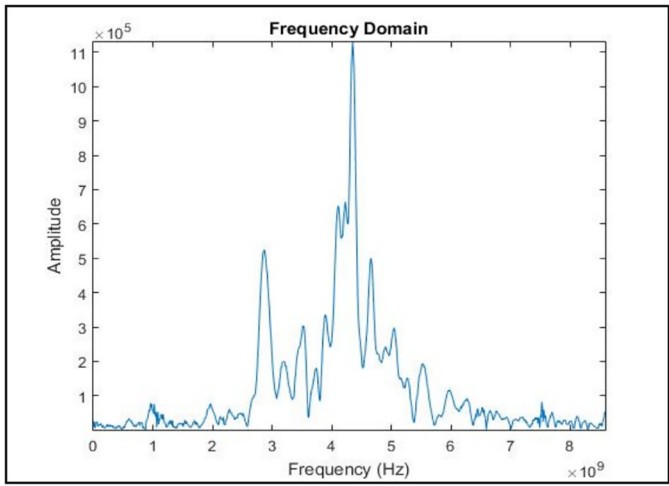

**Fig 6. The UWB signal in frequency domain.** Figure shows the representation of the signal in the frequency domain after the transformation.

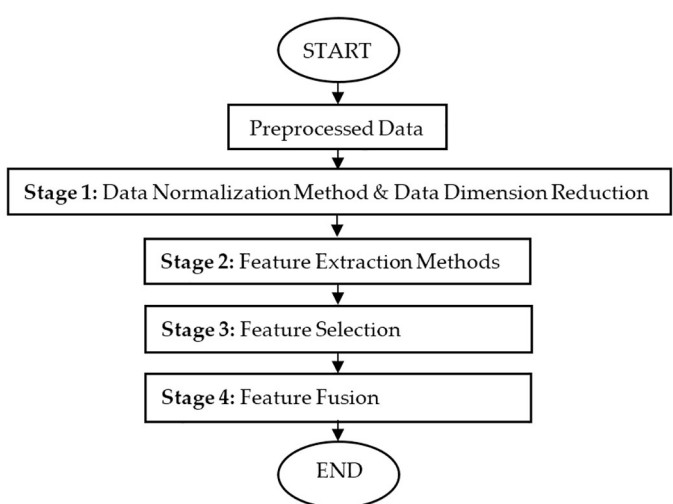

**Fig 7. Multi- stage feature selection method.** Figure shows the overall flow chart of the proposed MSFS method.

## Multi- stage feature selection method

Fig 7 illustrates the overall flow chart of the proposed MSFS method. It is divided into multiple stages [14]. Once the data is preprocessed, it is normalized to 10 different data normalization methods, and the data is reduced using PCA. Then, 10 different features will be extracted from each data normalization method. The best features are selected statistically from the extracted features based on the *p*- value and *F*- value. Then, the selected feature datasets are fused together to produce a newly proposed hybrid feature dataset which will be used for EBCD framework.

### Stage 1: Data normalization methods & data dimension reduction

Data normalization is a method to standardize the range of features without reducing the dimension of the data [5, 31–35]. Data normalization process is important since it is important to select the best features without eliminating useful information from the preprocessed data [31–35]. Conventional single stage feature selection having the drawback of possibly selecting data after eliminating useful data during feature extraction stage. Thus, for this work, raw data samples are normalized using ten different data normalization methods. Based on the comprehensive review done on the previous researches, five data normalization methods are chosen from the commonly used methods, namely, Decimal Scaling (DS), Z-score (ZS), Linear Scaling (LS), Min-Max (MM) and Mean & Standard Deviation (MSD) methods [31–35]. The other five data normalization methods are newly introduced in early breast cancer detection application, namely, Relative Logarithmic Sum Squared Voltage (RLSSV), Relative Logarithmic Voltage (RLV), Relative Voltage (RV), Fractional Voltage Change (FVC) and Relative Sum Squared Voltage (RSSV) [14, 36]. These data normalization methods are proposed by [14, 36] to overcome the baseline drift error that normally comes together with the data sample which affects the quality of the data samples. The received signals are in amplitude (V/m) versus time (ns) for time domain (refer Fig 5) and amplitude (V/m) vs frequency (Hz) for frequency domain (refer Fig 6). The amplitude (V/m) value is used as voltage input for these five data normalization methods.

Once the data is normalized, the normalized data is dimensionally reduced to remove redundant and statistically insignificant data [34]. The dimension of data is reduced as follows:

1. Each dataset has the maximum number of columns, c of principal components.

2. The last column of the principal component is reduced from the dataset. The remaining column is c-1.

3. The *p*- value is computed for the c-1 column of the principal components.

4. Step 2 and 3 are repeated until the *p*- value is less than 0.05.

5. The remaining number column of the principal components is recorded.

6. These processes are repeated for all ten new normalized datasets.

## Stage 2: Feature extraction methods

In order to perform the feature selection process in Stage 3 effectively, feature extraction method is applied on the 10 normalized datasets mentioned in the previous section. Ten features consist of combinations of statistical, time domain and frequency domain features are extracted from each normalized dataset. The features are Mean (M), Standard Deviation (SD), Skewness (S), Variance (V), Power Spectral Density (PSD), FFT Maximum Value (MAX), FFT Minimum Value (MIN), Independent Component Analysis (ICA), Shannon Entropy (SE) and Sure Entropy (SU) [27, 37].

Mean (M) is the ratio of the sum of values to the total number of values as shown in Eq (1). $v_1$ is the first value of data and $N$ is the data sample size.

$$M = \frac{v_1 + v_2 + v_3 + \dots v_N}{N} \tag{1}$$

Standard Deviation (SD) is used to measure the amount of variation of a set of values in a data as shown in Eq (2). $v$ is the first value of data and $N$ is the data sample size.

$$SD = \sqrt{\sum \frac{(v - \mu_N)^2}{N}} \tag{2}$$

Skewness (S) measures the asymmetry of distribution. Distribution is symmetry if it is looked same for both sides as shown in Eq (3). $v$ is the value of data, $N$ is the data sample size and $\mu_N$ is the mean.

$$S = \frac{\sum \frac{(v - \mu_N)^3}{N}}{\left( \sum \frac{(v - \mu_N)^2}{N} \right)^{\frac{3}{2}}} \tag{3}$$

Variance (V) measures how far the value is from the mean. It is measured using the Eq (4). $v$ is the value of data, $N$ is the data sample size and $\mu_N$ is the mean.

$$S = \sum \frac{(v - \mu_N)^2}{N} \tag{4}$$

Power spectral density (PSD) estimates the power in a different frequency range. The time domain data should be transformed into frequency domain data before further analysis. In this study, PSD is estimated using the Welch method which is defined as the moving window technique. Initially, FFT values are computed for each window and then, PSD values are calculated by averaging FFT values over all windows. The Hamming window function is used here because it has a good frequency resolution and reduces spectral leakage [37].

Maximum FFT (MAX) is the largest value in a set of data after the transformation of time domain data to frequency domain data using FFT. It is usually calculated using the max function in MATLAB. It is measured using Eq (5). $v_i$ is the first value of data, $v_j$ is the second value of data, $i$ is 1,2,3 . . . $i_n$, $j$ is 1,2,3 . . . $j_n$ and $N$ is the data sample.

$$MAX = v_i : v_i \geq v_j, i \neq j \ \forall \ i,j \in N \tag{5}$$

Minimum FFT (MIN) is the smallest value in a set of frequency domain data and is calculated using the min function in MATLAB. It is measured using the Eq (6). $v_i$ is the first value of data, $v_j$ is the second value of data, $i$ is 1,2,3 . . . $i_n$, $j$ is 1,2,3 . . . $j_n$ and $N$ is the data sample.

$$MIN = v_i : v_i \leq v_j, i \neq j \ \forall \ i,j \in N \tag{6}$$

Independant Component Analysis (ICA) identifies statistically independent values in a dataset. Eq (7) shows the statistical ICA model. $v_N$ is a set of observation vector, $S_N$ is a mixture of independent component vector, $N$ is the sample size and $A$ is $N^*N$ mixing mixture.

$$v_N = AS_N \tag{7}$$

Then, ICA finds the unmixing matrix W (inverse of A) to obtain the independent components (IC) as shown in Eq (8). $v_N$ is the observation vector and $N$ is the data sample size.

$$ICA = Wv_N \tag{8}$$

Entropy measures the uncertainty distribution and complexity characteristics in data. Shannon entropy (SE) is defined as shown in Eq (9) which describe the internal characteristic information in a data. $v_i$ is the value of data.

$$SE = -\sum_i v_i \ log(v_i) \tag{9}$$

Sure entropy (SU) is the measurement of surface entropy and is defined as shown in Eq (10). $S(N)$ is sure entropy, $v$ is the value of data, $N$ is the data sample size and $\varepsilon$ is the positive threshold which is determined using Steins unbiased risk estimate principle.

$$|v| \leq \varepsilon \rightarrow SU = -\sum_N min(v^2 \ \varepsilon^2) \tag{10}$$

## Stage 3: Feature selection

Stage 3 is divided into two analyses. The first analysis is on selection of normalization method. The second analysis is on selection of features. Both analyses are conducted using statistical computations of statistical $p$- value and $F$- value [14, 36].

**Analysis 1: Selection of normalization methods.** 10 features are extracted from each data normalization method, which resulted total of 100 extracted features. Each normalized method has a data matrix of [6750 x 10] where, 6750 is the number of data samples and 10 is the number of features as in Eqs (11) to (20). Each data matrix will be referred with name [DS], [ZS],

[LS], [MM], [MSD], [RLSSV], [RLV], [RV], [FVC] and [RSSV] respectively.

$$[DS] = [SD_{DS}, M_{DS}, V_{DS}, S_{DS}, SE_{DS},$$
$$ICA_{DS}, SU_{DS}, PSD_{DS}, MAX_{DS}, MIN_{DS}] \tag{11}$$

$$[ZS] = [SD_{ZS}, M_{ZS}, V_{ZS}, S_{ZS}, SE_{ZS}, ICA_{ZS}, SU_{ZS}, PSD_{ZS}, MAX_{ZS}, MIN_{ZS}] \tag{12}$$

$$[LS] = [SD_{LS}, M_{LS}, V_{LS}, S_{LS}, SE_{LS}, ICA_{LS}, SU_{LS}, PSD_{LS}, MAX_{LS}, MIN_{LS}] \tag{13}$$

$$[MM] = [SD_{MM}, M_{MM}, V_{MM}, S_{MM}, SE_{MM},$$
$$ICA_{MM}, SU_{MM}, PSD_{MM}, MAX_{MM}, MIN_{MM}] \tag{14}$$

$$[MSD] = [SD_{MSD}, M_{MSD}, V_{MSD}, S_{MSD}, SE_{MSD},$$
$$ICA_{MSD}, SU_{MSD}, PSD_{MSD}, MAX_{MSD}, MIN_{MSD}] \tag{15}$$

$$[RLSSV] = [SD_{RLSSV}, M_{RLSSV}, V_{RLSSV}, S_{RLSSV}, SE_{RLSSV},$$
$$ICA_{RLSSV}, SU_{RLSSV}, PSD_{RLSSV}, MAX_{RLSSV}, MIN_{RLSSV}] \tag{16}$$

$$[RLV] = [SD_{RLV}, M_{RLV}, V_{RLV}, S_{RLV}, SE_{RLV},$$
$$ICA_{RLV}, SU_{RLV}, PSD_{RLV}, MAX_{RLV}, MIN_{RLV}] \tag{17}$$

$$[RV] = [SD_{RV}, M_{RV}, V_{RV}, S_{RV}, SE_{RV},$$
$$ICA_{RV}, SU_{RV}, PSD_{RV}, MAX_{RV}, MIN_{RV}] \tag{18}$$

$$[FVC] = [SD_{FVC}, M_{FVC}, V_{FVC}, S_{FVC}, SE_{FVC},$$
$$ICA_{FVC}, SU_{FVC}, PSD_{FVC}, MAX_{FVC}, MIN_{FVC}] \tag{19}$$

$$[RSSV] = [SD_{RSSV}, M_{RSSV}, V_{RSSV}, S_{RSSV}, SE_{RSSV},$$
$$ICA_{RSSV}, SU_{RSSV}, PSD_{RSSV}, MAX_{RSSV}, MIN_{RSSV}] \tag{20}$$

Statistical tests are conducted on each data matrix in Eqs (11) to (20) to find the $p$- value and the $F$- value of the respective data matrix. Table 3 shows the $p$- value and $F$- value of the

**Table 3. $p$- value and $F$- value computation for the data matrices.**

| Data Matrices | $p$- value | $F$- value |
|---|---|---|
| [DS] | 0.023971 | 1.07541 |
| [ZS] | 0.022969 | 1.075852 |
| [LS] | 1 | 0.59923 |
| [MM] | 1 | 0.608231 |
| [MSD] | 0.049447 | 1.062474 |
| [RLSSV] | 0.013245 | 47.85685 |
| [RLV] | 0.494978 | 1.00003 |
| [RV] | 0.029273 | 1.071593 |
| [FVC] | 0.025485 | 1.074304 |
| [RSSV] | 0.046762 | 1.063453 |

Table shows the $p$- value and $F$- value of the ten data matrix.

**Table 4.** *F-* value of extracted features for finalized normalization method.

| | Data Normalization Method | | | | |
| | ZS | DS | RLSSV | RV | FVC |
|---|---|---|---|---|---|
| *SD* | - | 55.12 | 10.95 | 103.64 | 160.47 |
| *M* | - | - | 5.46 | - | - |
| *V* | - | 53.79 | 13.76 | 88.70 | 135.12 |
| *S* | - | - | 126.54 | - | - |
| *SE* | - | 70.59 | 9.48 | 154.30 | 140.44 |
| *ICA* | 8.48 | 2.36 | - | 10.22 | - |
| *SU* | 282.76 | 53.89 | 3.47 | 88.70 | 48.21 |
| *PSD* | 222.52 | - | - | 11.59 | 3.29 |
| *MAX* | 6.76 | 4.26 | - | - | - |
| *MIN* | 4.45 | 37.55 | - | - | - |

Table shows the result of the analysis.

ten data matrix. The five data matrix which pass the selection criteria of having *p-* value less than 0.05 and highest *F-* value are selected. Based on Table 3, it can be seen that data matrices [LS], [MM] and [RLV] did not meet the first selection criterion of ($p < 0.05$), and thus, rejected. The second selection criterion (highest *F-* value) is checked for the remaining seven data matrices, and it is found that [RLSSV], [DS], [ZS], [RV] and [FVC] are selected based on the two mentioned selection criteria. These five data matrices are used for further analysis.

**Analysis 2: Selection of feature extraction methods.**   The features mentioned in Eqs (1) to (10) are extracted using the finalized data normalization methods in Analysis 1. The 50 extracted features undergo the same selection criteria of *p-* value is less than 0.05 and highest *F-* value. If the feature is statistically significant (*p-* value is less than 0.05), then features are selected for second criterion check. Among the statistically significant features, ten best features are selected based on highest *F-* value. Table 4 shows the result of the analysis. The statistically insignificant features (*p-* value > 0.05) are represented with symbol (-), and therefore, are removed from the feature selection listing. Out of the remaining features, ten features with highest *F-* values (highlighted in the table) are selected for Stage 4. Table 5 shows the ranking of the selected features in Stage 3 based on the *F-* values (in descending order) for the respective normalization methods [38].

**Table 5. Ranking of selected features in Stage 3 based on the *F-* value.**

| Feature | Data Normalization Method | *F-* value | Dataset Name |
|---|---|---|---|
| *SU* | ZS | 282.76 | [SU-ZS] |
| *PSD* | ZS | 222.52 | [PSD-ZS] |
| *SD* | FVC | 160.47 | [SD-FVC] |
| *SE* | RV | 154.30 | [SE-RV] |
| *SE* | FVC | 140.44 | [SE-FVC] |
| *V* | FVC | 135.12 | [V-FVC] |
| *S* | RLSSV | 126.55 | [S-RLSSV] |
| *SD* | RV | 103.64 | [SD-RV] |
| *V* | RV | 88.70 | [V-RV] |
| *SU* | RV | 88.70 | [SU-RV] |

Table shows the ranking of the selected features in Stage 3 based on the *F-* values (in descending order).

## Stage 4: Feature fusion

Feature fusion is the hybridization of the statistically selected features. Table 5 shows the ranking of the selected features in Stage 3 based on the *F*- values (in descending order). They are used in the further analyses using the Dataset Name assigned.

In this stage, the selected features are fused together to develop the proposed hybrid feature. Each dataset is reduced to single column [6750 rows x 1 column] using Singular Value Decomposition (SVD) method [14]. Before data fusion, each dataset is having [6750 x 1] dimension as shown in Fig 8. The ten individual datasets are fused together to form the proposed hybrid feature with dimension [6750 x 10]. First column to the last column are assigned based on the ranking in Table 5, starting from [SU-ZS] to [SU-RV].

*F*- value is computed for different number of hybrid features, starting from fusion of 10 best features, decreasing until fusion of 2 best features. The results are tabulated in Table 6. From the result, it can be seen that fusion of 6 to 10 best features are giving best result range in terms of *F*- value. Fusion of 8 best features recorded the highest F- value. Three datasets with highest F- value are chosen for further analysis. The three datasets are referred to as 10-HybridFeature, 9-HybridFeature and 8-HybridFeature datasets throughout the paper.

The overall block diagram of the proposed MSFS method is shown in Figs 9 and 10. Fig 9 shows the data normalization selection and data dimension reduction, while Fig 10 shows the feature extraction selection, feature fusion and formulation of the proposed feature datasets. The dimension (Rows x Columns) of the datasets is shown in square brackets ([]).

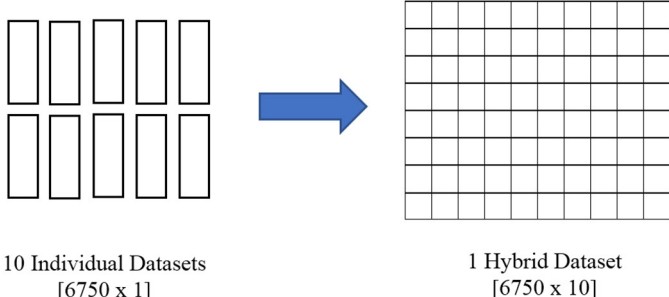

10 Individual Datasets
[6750 x 1]

1 Hybrid Dataset
[6750 x 10]

**Fig 8. Fusion process of the hybrid feature.** Figure shows the [6750 x 1] datasets are fused together to form the proposed hybrid feature with dimension [6750 x 10].

**Table 6.** *F*- value of the Datasets with Different Number of Features.

| Number of Hybrid Features | Dimension[Rows x Columns] | *F*- value | Dataset Name |
|---|---|---|---|
| 10 | [135 x 10] | 938267.28 | 10-HybridFeature |
| 9 | [135 x 9] | 939582.08 | 9-HybridFeature |
| 8 | [135 x 8] | 942111.32 | 8-HybridFeature |
| 7 | [135 x 7] | 935645.64 | - |
| 6 | [135 x 6] | 937341.03 | - |
| 5 | [135 x 5] | 229.30 | - |
| 4 | [135 x 4] | 208.70 | - |
| 3 | [135 x 3] | 168.38 | - |
| 2 | [135 x 2] | 702.17 | - |

Table shows the *F*- value of the Datasets with Different Number of Features.

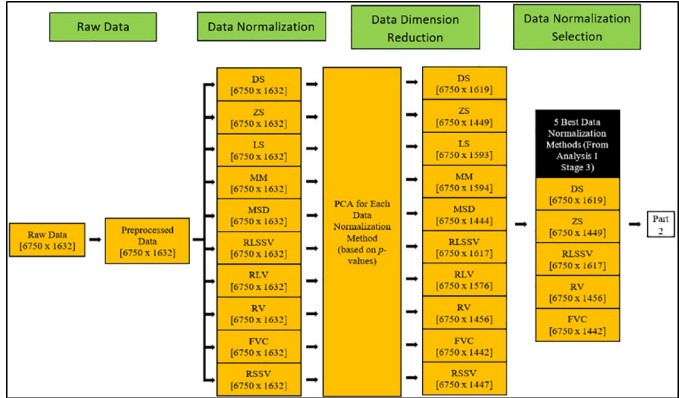

**Fig 9. Block diagram of proposed MSFS method (Part 1).** Figure shows shows the data normalization selection and data dimension reduction.

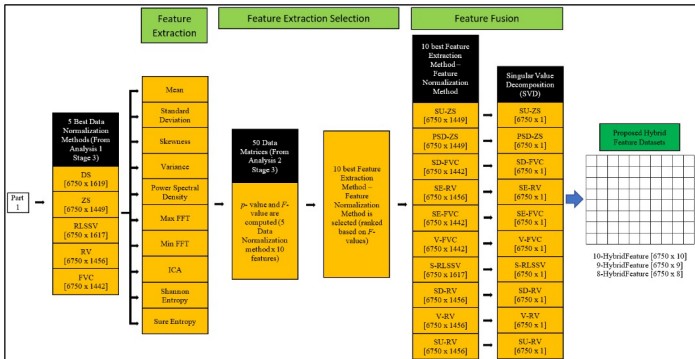

**Fig 10. Block diagram of proposed MSFS method (Part 2).** Figure shows the feature extraction selection, feature fusion and formulation of the proposed feature datasets.

## Classification of breast cancer size

For breast cancer size classification tests with classifiers, the three hybrid feature datasets are used. The SU-ZS dataset from Stage 4 which recorded the highest *F*- value as an individual feature extraction method- feature normalization method is added as a comparison dataset. Three commonly used machine learning methods, namely, Support Vector Machine (SVM), Probabilistic Neural Network (PNN) and Naïve Bayes (NB) classifiers are used for breast cancer size classification [19, 39–42].

The classifier parameters are set in such a way that, for SVM, the linear kernel function is used. For PNN, spread factor of 0.1 and four layers (input, pattern, summation and output layers) are used. There is no classifier parameter for NB since it does not require tuning parameters [40]. There are six possible outputs, which are the five breast cancer tumor sizes (2 mm, 3 mm, 4 mm, 5 mm, and 6 mm) and non- existence of the tumor. 750 non- existence of tumor samples were added to the sample size. Two processes are involved in the classification: training and testing. The training and testing are conducted using k- fold cross-validation method [43]. Fig 11 shows the k- fold cross- validation method used in this research [43–45]. Ten k- folds are used. The total number of 7500 data samples (6750 samples with tumor + 750 samples with non- existence of tumor) from each dataset are divided into 10 equal portions (folds).

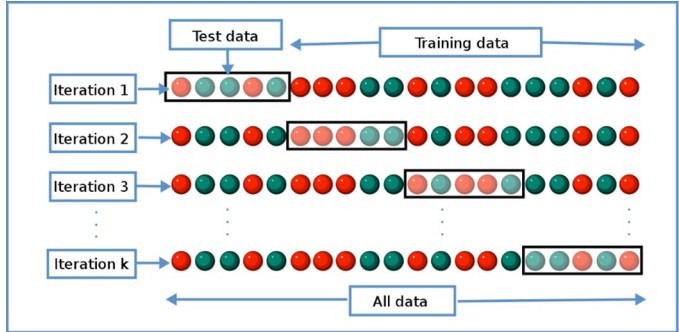

**Fig 11. k- Fold Cross-validation method** [44]. Figure shows the k- fold cross- validation method used in this research.

Each fold will have 750 data samples. The training process is done using the 9 folds data, while the testing is done using the remaining a fold data. Each fold will take turn to be the testing fold, until the training- testing process completed. Confusion matrices are generated for each iteration, and the accuracy, sensitivity and specificity are calculated for each iteration using Eqs (21) to (23). The average classification accuracy, sensitivity and specificity of all folds are considered as the performance of the classifier [14]. TP is true positive (indicates correct classification of cancer size), FP is false positive (indicates incorrect classification of cancer size), TN is true negative (indicates correct classification of non- existence events), and FN is false negative (indicates incorrect classification of non- existence events) of breast cancer size prediction. Calculating the accuracy, specificity and sensitivity are important as to have a successful early detection of breast cancer and to reduce misclassification in the classification. Medically, when a breast is tested for lesions at early stage, there are possible chances of high misclassification to happen. Misclassification can be considered as the event where lesion is available, but not detected by the system, or no lesion available but the classifier detects a tumor. Having such possibility will definitely affect the overall efficiency of the system, and thus, must be eliminated or reduced.

$$\text{Accuracy} = \frac{TN + TP}{TN + TP + FN + FP} \tag{21}$$

$$\text{Sensitivity} = \frac{TP}{TP + FN} \tag{22}$$

$$\text{Specificity} = \frac{TN}{TN + FP} \tag{23}$$

## Results

Fig 12 shows the performance of SVM, NB and PNN for SU-ZS, 8-HybridFeature, 9-Hybrid-Feature and 10-HybridFeature datasets. For SU-ZS dataset, the accuracies for SVM, NB and PNN are recorded as 84.39%, 83.69% and 82.97% respectively. These accuracies are considered as the benchmarking accuracies of this study to show the effectiveness of the proposed method. Table 7 shows the performance of SVM, NB and PNN for SU-ZS Dataset (Reference Dataset). 8-HybridFeature dataset records the highest classification performance, whereas, 10-Hybrid-Feature dataset has the lowest classification performance of all classifiers.

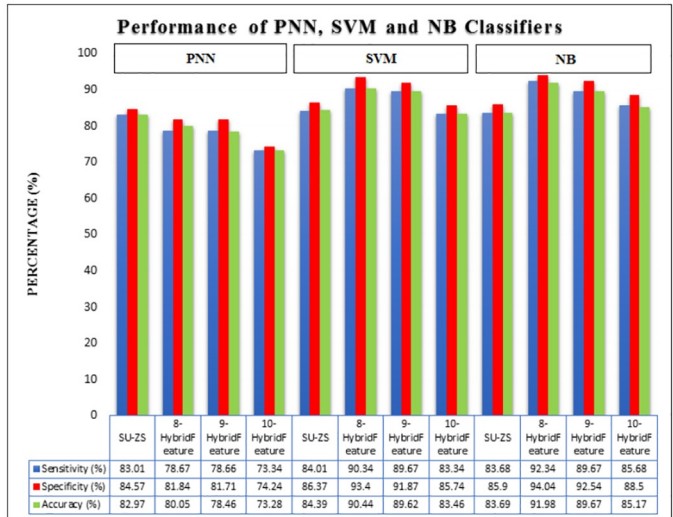

**Fig 12. Performance of SVM, NB and PNN for SU-ZS, 8-HybridFeature, 9-HybridFeature and 10-HybridFeature Datasets.** Figure shows the performance of SVM, NB and PNN for SU-ZS, 8-HybridFeature, 9-HybridFeature and 10-HybridFeature datasets.

For PNN classifier, the highest result is achieved by 8-HybridFeature dataset by obtaining 80.05%, 78.67%, 81.84% for accuracy, sensitivity and specificity respectively. The lowest result is achieved by 10-HybridFeature dataset by obtaining 73.28% accuracy, 73.34% sensitivity and 74.24% specificity.

For SVM classifier, 8-HybridFeature dataset obtains the highest classification performance, which is 90.44%, 90.34% and 93.40% for accuracy, sensitivity and specificity respectively, while the 10-HybridFeature dataset obtains the lowest classification performance with recorded performance 83.46%, 83.34% and 85.74% for accuracy, sensitivity and specificity respectively.

For NB classifier, 8-HybridFeature dataset achieves 91.98%, 92.34% and 94.04% for accuracy, sensitivity and specificity (the highest result). The 10-HybridFeature dataset achieves 85.17%, 85.68% and 88.50% for accuracy, sensitivity and specificity respectively. In general, the classifiers are able to classify with lowest misclassification rate. The three hybrid feature datasets also recorded a better result in general compared to individual features (Table 7) except for PNN classifier. It can be due to unoptimized spread factor for PNN classifier.

8-HybridFeature dataset is proven to be having the best result compared to other datasets because it contains very strong and significant features. It is proven by validation using statistical approach ($p$- value and $F$- value) and machine learning approach. 8-HybridFeature dataset has improved the classification process to be more than 90% accurate for SVM and NB classifiers. Therefore, it can be concluded that the selected hybrid features through MSFS process are able to improve the overall classifier performance.

**Table 7. Performance of SVM, NB and PNN for SU-ZS dataset (Reference dataset).**

|  | Accuracy | | | | Sensitivity | | | | Specificity | | | |
|---|---|---|---|---|---|---|---|---|---|---|---|---|
|  | SVM | NB | PNN | Mean | SVM | NB | PNN | Mean | SVM | NB | PNN | Mean |
| *SU—ZS* | 84.39 | 83.69 | 82.97 | 83.68 | 84.01 | 83.68 | 83.01 | 83.57 | 86.37 | 85.90 | 84.57 | 85.61 |

Table shows the performance of SVM, NB and PNN for SU-ZS Dataset (Reference Dataset).

**Table 8. Comparison with previous researches (Data from these researches were fed into the proposed method in this paper).**

| Researcher | [27] | [19] | Proposed Work |
|---|---|---|---|
| *Preprocessing Method* | Extract statistical features | PCA | MSFS |
| *Features* | 4 Features | First 450 PCs | 8 Features |
| *Machine Learning* | FFNN | FFNN | Naïve Bayes |
| *Training/Validation/Testing Ratio* | 70/15/15 | 70/15/15 | K-fold Cross Validation |
| *Accuracy* (%) | 97.51 | 95.80 | 98.17 |

The dataset from [19, 27] were used in the proposed method for this analysis.

**Table 9. Comparison with previous researches (Data from this research is fed into the the existing methods).**

| Researcher | [27] | [19] | Proposed Work |
|---|---|---|---|
| *Preprocessing Method* | Extract statistical features | PCA | MSFS |
| *Features* | 4 Features | First 450 PCs | 8 Features |
| *Machine Learning* | FFNN | FFNN | Naïve Bayes |
| *Training/Validation/Testing Ratio* | 70/15/15 | 70/15/15 | K-fold Cross Validation |
| *Accuracy* (%) | 82.61 | 74.15 | 94.07 |

The dataset from this proposed research is used for this analysis in the existing methods from [19, 27].

The accuracy of the proposed method is compared with the existing work, [1, 42] as demonstrated in Tables 8 and 9. Two analyses were performed to test the efficiency of the dataset and proposed method. For the first analysis (presented in Table 8), the data from the existing researches were fed into the proposed method in this paper. For the second analysis (presented in Table 9, the data from this research was fed into the existing methods. The result proves the proposed MSFS method and hybrid feature are better compared to the other existing method. The proposed MSFS method achieves 91.98% which is much better compared to the previous method (74.61%). It improves approximately 17% of accuracy in comparison to the of the previous study.

## Early breast cancer detection (EBCD) framework

The EBCD framework is a user- friendly platform developed to assist medical personnel on early breast cancer detection. The complete EBCD framework consists of the integration of software and hardware (configuration, obtain data sample/s, analogue to digital conversion and scan data saving), preprocessing, detection of breast cancer size and visualization of the output in the 2D and 3D environment. The presented Graphical User Interface (GUI) in 2D and 3D environment are combination of few research arms on early breast cancer detection, consist of cancer existence, cancer size detection (covered in this paper), location detection, and cancer type detection. Cancer existence, location detection and cancer type detection is presented in other papers from the authors [18]. Once the framework is developed, the framework is converted into a standalone executable file (.exe file) in order to make the system flexible and easy to access.

EBCD framework is developed using MSFS method and NB for breast cancer size detection as shown in Fig 13. The fresh data sample (new data sample) will not go through MSFS processes from scratch again (Stage 1 to Stage 4) as the data normalization methods and features extraction methods involved in forming 8-HybridFeature are already selected and identified

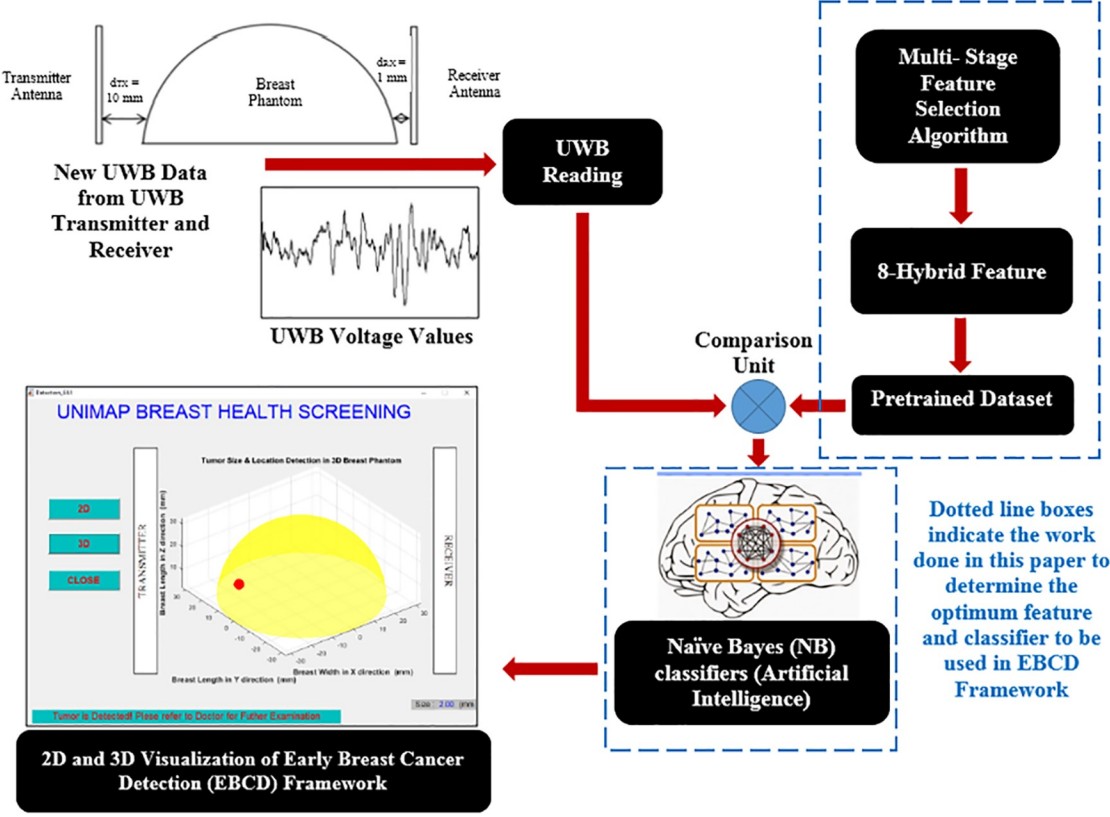

**Fig 13. Block diagram of EBCD framework developed using MSFS method and NB classifier for early breast cancer detection.**
Figure shows the block diagram of EBCD Framework developed using MSFS method and NB classifier for Early Breast Cancer
Detection.

through this work. Thus, it helps in reducing the time consumption and computational complexity in EBCD framework. The proposed EBCD algorithm is implemented in the UWB system to develop a complete early breast cancer size detection framework. Figs 14 and 15 show the example of visualization layout in the 2D and 3D environment to detect 2 mm tumor at the location of 2.5 mm, 32.5 mm and 50 mm for x, y and z coordinates respectively.

## Conclusions

MSFS method is proposed for early breast cancer detection application. The proposed MSFS method has four stages. The first stage consists of data normalization methods and data reduction. The second stage consist of feature extraction methods, while third stage and fourth stage consist of feature selection and feature fusion, respectively. The selection of data normalization methods and features are done by computing the $p$- value and $F$- value in each stage. The raw data samples go through these stages in order to identify the best data normalization methods, best feature extraction methods and optimum hybrid features. The hybrid features are fused together using feature fusion technique.

Three different datasets (8-HybridFeature, 9-HybridFeature and 10-HybridFeature) are formed through MSFS method. These datasets are tested using three different supervised classifiers (SVM, NB and PNN) to check the robustness of the features. All classifiers obtain classification accuracy of more than 70%. The highest classification accuracy is obtained by

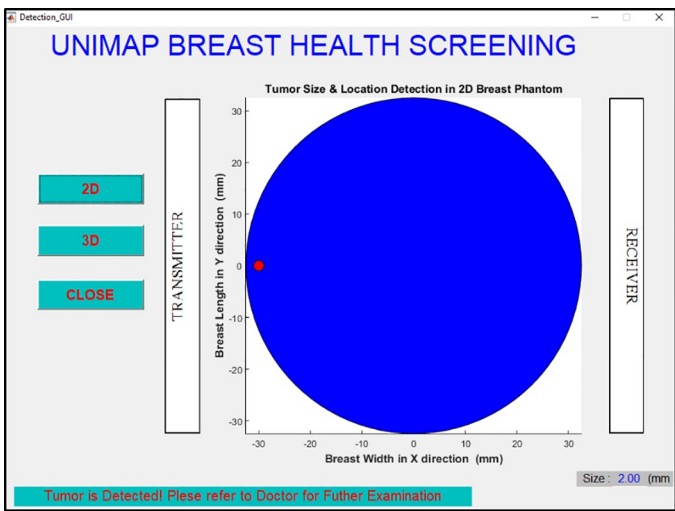

**Fig 14. Visualization of detected 2 mm tumor size in 2D environment.** Figure shows the example of visualization layout in the 2D environment to detect 2 mm tumor at the location of 2.5 mm, 32.5 mm and 50 mm for x, y and z coordinates.

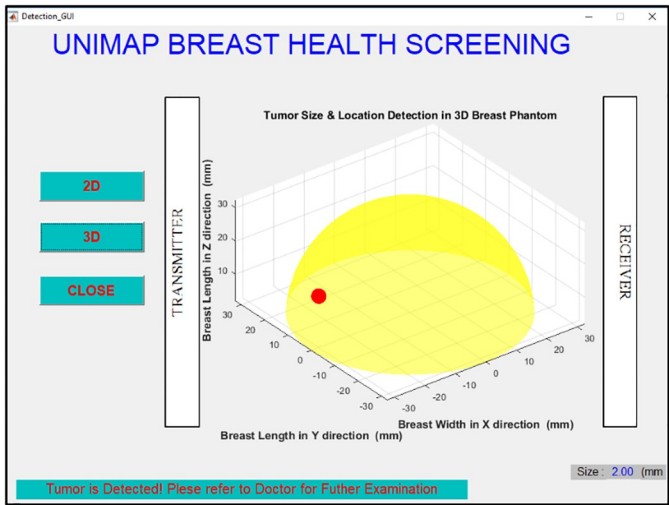

**Fig 15. Visualization of detected 2 mm tumor size in 3D environment.** Figure shows the example of visualization layout in the 3D environment to detect 2 mm tumor at the location of 2.5 mm, 32.5 mm and 50 mm for x, y and z coordinates.

8-HybridFeature dataset tested in NB classifier (91.98%). A complete early breast cancer detection framework is developed. The finalized MSFS methods are implemented in the EBCD framework. The detected size is visualized in the 2D and 3D environment.

## Acknowledgments

The authors would like to express their gratitude to Universiti Malaysia Perlis (UniMAP) and Universiti Malaysia Pahang (UMP) for the extensive support in providing research facilities for this research.

## Author Contributions

**Conceptualization:** V. Vijayasarveswari, A. M. Andrew, M. Jusoh, S. Khatun.

**Data curation:** V. Vijayasarveswari.

**Formal analysis:** V. Vijayasarveswari, T. Sabapathy, R. A. A. Raof, S. Khatun.

**Funding acquisition:** M. Jusoh, H. A. Rahim.

**Investigation:** V. Vijayasarveswari, A. M. Andrew, T. Sabapathy, R. A. A. Raof, M. N. M. Yasin, S. Khatun.

**Methodology:** A. M. Andrew, M. Jusoh, T. Sabapathy, R. A. A. Raof, S. Khatun.

**Resources:** M. Jusoh, M. N. M. Yasin, R. B. Ahmad.

**Supervision:** A. M. Andrew, M. Jusoh, M. N. M. Yasin, R. B. Ahmad.

**Validation:** V. Vijayasarveswari, M. Jusoh, T. Sabapathy, R. A. A. Raof, S. Khatun, H. A. Rahim.

**Visualization:** V. Vijayasarveswari, M. Jusoh, R. A. A. Raof, S. Khatun.

**Writing – original draft:** V. Vijayasarveswari, A. M. Andrew.

**Writing – review & editing:** V. Vijayasarveswari, A. M. Andrew, H. A. Rahim.

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
