## [Decision Letter · Decision Letter 0]

5 Mar 2020

PONE-D-20-02624

Multi- Stage Feature Selection (MSFS) Algorithm for UWB- Based Early Breast Cancer Size Prediction

PLOS ONE

Dear DR ANDREW,

Thank you for submitting your manuscript to PLOS ONE. After careful consideration, we feel that it has merit but does not fully meet PLOS ONE’s publication criteria as it currently stands. Therefore, we invite you to submit a revised version of the manuscript that addresses the points raised during the review process.

We would appreciate receiving your revised manuscript by Apr 19 2020 11:59PM. To enhance the reproducibility of your results, we recommend that if applicable you deposit your laboratory protocols in protocols.io, where a protocol can be assigned its own identifier (DOI) such that it can be cited independently in the future. For instructions see: http://journals.plos.org/plosone/s/submission-guidelines#loc-laboratory-protocols

We look forward to receiving your revised manuscript.

Kind regards,

Muhammad Zubair

Academic Editor

PLOS ONE

Additional Editor Comments (if provided):

This manuscript has been reviewed by three of our reviewers. Please see the comments below.

Overall the criticism indicates that the present draft needs substantial revision. However, it seems that a suitably revised version would merit further attention. If you choose to resubmit, be sure to address each criticism in detail.

2) Please upload a copy of Figure 9 to which you refer in your text on page xx. If the figure is no longer to be included as part of the submission please remove all reference to it within the text.

3) Please include your tables as part of your main manuscript and remove the individual files. Please note that supplementary tables (should remain/ be uploaded) as separate "supporting information" files

4) Please update your submission to use the PLOS LaTeX template. The template and more information on our requirements for LaTeX submissions can be found at http://journals.plos.org/plosone/s/latex.

Reviewers' comments:

Reviewer's Responses to Questions

**Comments to the Author**

1. Is the manuscript technically sound, and do the data support the conclusions?

Reviewer #1: No

Reviewer #2: Yes

Reviewer #3: Partly

2. Has the statistical analysis been performed appropriately and rigorously? 

Reviewer #1: No

Reviewer #2: I Don't Know

Reviewer #3: I Don't Know

3. Have the authors made all data underlying the findings in their manuscript fully available?

Reviewer #1: Yes

Reviewer #2: Yes

Reviewer #3: Yes

4. Is the manuscript presented in an intelligible fashion and written in standard English?

Reviewer #1: Yes

Reviewer #2: Yes

Reviewer #3: Yes

5. Review Comments to the Author

Reviewer #1: The paper is interesting, but there is a mayor concern that should be addressed.

Usually sensitivity/specificity are defined following these assumptions:

negative cases: breast with NO lesions

positive: breast with lesions

Do you consider also breast with NO lesions in your experiments?

how many data of breast with NO lesions are you considering?

If you are not considering breast with NO lesions, how can you calculate specificity?

A deeper description/investigation/discussion of sensitivity, specificity and accuracy is required, especially using a medical point of view.

Reviewer #2: The authors have explained well, the idea and main steps necessary to support their paper. The framework highlights an interesting aspect of early breast cancer detection systems. However, I felt that while reading the manuscript their is lack of further explanation, especially the part where the cancer detection method is not explained well. The statistical procedure is explained, however is not supported by some real data. A further explanation with support from some read data, will certainly increase the overall impact of the manuscript.

Reviewer #3: Interesting work. A good approach to solve the problem in breast cancer detection.

I'd have some questions:

- What do you mean by "researchers used either real-time machines (...) or machine learning to analyze UWB signals" while citing ref [1]?

- In the phantom section. Glass is giving a proper shape to the phantom, but is material also mimicking the skin dielectric properties? Maybe call it skin is not exact.

- Any details on used antenna?

- Are the antennas touching the phantom?

- Antennas are placed in ony one position. Have the authors thought on a multi-view approach?

- Did the authors compare their proposed method to a known one using the same dataset? (Maybe is already answered in Table 6). My question is if same dataset is used on both cases.

- If I got correctly. Classifier is giving information on size only. Which is the idea in giving a 3D image of this if position is not known? Is it to give an idea on relative size to the breast?

- Did the authors try with a case that is out of the ones already defined? I mean: What happens if a testing data with a tumour of a different size is used? (let's say 2.5 mm or 1 mm)

6. PLOS authors have the option to publish the peer review history of their article (what does this mean?). If published, this will include your full peer review and any attached files.

Reviewer #1: No

Reviewer #2: No

Reviewer #3: No

---

## [Author Response · Author response to Decision Letter 0]

13 May 2020

All the review and comments from Academic Editor, Reviewer 1, Reviewer 2 and Reviewer 3 are completed and attached in the table with rebuttal letter. All comments are carefully amended by the authors of the paper. Thank you so much!

---

## [Decision Letter · Decision Letter 1]

8 Jun 2020

PONE-D-20-02624R1

Multi- Stage Feature Selection (MSFS) Algorithm for UWB- Based Early Breast Cancer Size Prediction

PLOS ONE

Dear Dr. ANDREW,

Thank you for submitting your manuscript to PLOS ONE. After careful consideration, we feel that it has merit but does not fully meet PLOS ONE’s publication criteria as it currently stands. Therefore, we invite you to submit a revised version of the manuscript that addresses the points raised during the review process.

We look forward to receiving your revised manuscript.

Kind regards,

Muhammad Zubair

Academic Editor

PLOS ONE

Additional Editor Comments (if provided):

The manuscript has been improved. However, there are still some major points that should be addressed. Please work on them and revise the manuscript accordingly.

Reviewers' comments:

**Reviewer #1: **Please update the ref, including for example:

Aldhaeebi, M.A.; Alzoubi, K.; Almoneef, T.S.; Bamatraf, S.M.; Attia, H.; Ramahi, O.M. Review of Microwaves Techniques for Breast Cancer Detection. Sensors 2020, 20, 2390.

Rana, S.P., Dey, M., Tiberi, G. et al. Machine Learning Approaches for Automated Lesion Detection in Microwave Breast Imaging Clinical Data. Sci Rep 9, 10510 (2019). https://doi.org/10.1038/s41598-019-46974-3

**Reviewer #3**: Thanks to the authors for their answers.

I have still a couple of comments on some issues:

1. Authors declare in the response: "it is statistically validated that the material dielectric properties are same as the skin.". But Glass has, as declared in table 1, an epsilon_r between 3.5 and 10, while sigma value is considered negligible. These are normal values for glass. When we check skin values in [10], at the central frequency of 4.3 GHz epsilon_r is greater than 30 and sigma is almost 3 S/m. These values are confirmed in http://niremf.ifac.cnr.it/tissprop/

(eps= 36.342 and sigma=2.5443 S/m for dry skin).

2. Antenna Details. In Table 2 seems that Patch Description and Substrate Description contain the same information.

I expected a scheme, photo or drawing of the antenna, or a S11 plot in the frequency range of interest.

3. My question related to the comparison with a different algorithm was oriented to give a better glance at how your method works compared to existing and validated methods.

The other questions were answered in the authors' response.

Thank you,

---

## [Author Response · Author response to Decision Letter 1]

23 Jul 2020

All the mentioned/ suggested comments are improved in the revised paper. The details are explained further in the Response to Reviewer. Thank you.

---

## [Editor Report · Decision Letter 2]

29 Jul 2020

Multi- Stage Feature Selection (MSFS) Algorithm for UWB- Based Early Breast Cancer Size Prediction

PONE-D-20-02624R2

Dear Dr. ANDREW,

We’re pleased to inform you that your manuscript has been judged scientifically suitable for publication and will be formally accepted for publication once it meets all outstanding technical requirements.

Kind regards,

Muhammad Zubair

Academic Editor

PLOS ONE
---

## [Editor Report · Acceptance letter]

3 Aug 2020

PONE-D-20-02624R2 

Multi- stage feature selection (MSFS) algorithm for UWB-based early breast cancer size prediction 

Dear Dr. Andrew:

I'm pleased to inform you that your manuscript has been deemed suitable for publication in PLOS ONE. Congratulations! Your manuscript is now with our production department. 

Kind regards, 

on behalf of

Dr. Muhammad Zubair 

Academic Editor

PLOS ONE